**Data Availability Statement:** All relevant data are within the manuscript. Additional data are available

# Clinical characteristics of Japanese patients with chronic obstructive pulmonary disease (COPD) with comorbid interstitial lung abnormalities: A cross-sectional study

Manabu Ono[1]*, Seiichi Kobayashi[1], Masakazu Hanagama[1‡], Masatsugu Ishida[1‡], Hikari Sato[1‡], Tomonori Makiguchi[2‡], Masaru Yanai[1]

1 Department of Respiratory Medicine, Japanese Red Cross Ishinomaki Hospital, Ishinomaki, Miyagi, Japan,
2 Department of Respiratory Medicine, Hirosaki University Graduate School of Medicine, Hirosaki, Japan

☯ These authors contributed equally to this work.
‡ These authors also contributed equally to this work.
* ono_manabu@rm.med.tohoku.ac.jp

## Abstract

Smoking-related interstitial lung abnormalities are different from specific forms of fibrosing lung disease which might be associated with poor prognoses. Chronic obstructive pulmonary disease with comorbid interstitial lung abnormalities and that with pulmonary fibrosis are considered different diseases; however, they could share a common spectrum. We aimed to evaluate the clinical characteristics of Japanese patients with chronic obstructive pulmonary disease and comorbid interstitial lung abnormalities. In this prospective observational study, we analyzed data from the Ishinomaki COPD Network Registry. We evaluated the clinical characteristics of patients with chronic obstructive pulmonary disease with and without comorbid interstitial lung abnormalities by comparing the annualized rate of chronic obstructive pulmonary disease exacerbations per patient during the observational period. Among 463 patients with chronic obstructive pulmonary disease, 30 (6.5%) developed new interstitial lung abnormalities during the observational period. After 1-to-3 propensity score matching, we found that the annualized rate of chronic obstructive pulmonary disease exacerbations per patient during the observational period was 0.06 and 0.23 per year in the interstitial lung abnormality and control groups, respectively (P = 0.043). Our findings indicate slow progression of interstitial lung abnormality lesions in patients with pre-existing chronic obstructive pulmonary disease. Further, interstitial lung abnormality development did not significantly influence on chronic obstructive pulmonary disease exacerbation. We speculate that post-chronic obstructive pulmonary disease interstitial lung abnormalities might involve smoking-related interstitial fibrosis, which is different from specific forms of fibrosing lung disease associated with poor prognoses.

from the Japanese Red Cross Ishinomaki Hospital Institutional Data Access / Ethics Committee (contact via the Japanese Red Cross Ishinomaki Hospital) for researchers who meet the criteria for access to confidential data. Data access requests may be made to the data access committee (E-mail: renkei@ishinomaki.jrc.or.jp) of the Ishinomaki COPD Network (ICON) Registry.

**Funding:** The authors received no specific funding for this work.

**Competing interests:** The authors have declared that no competing interests exist.

## Introduction

Chronic obstructive pulmonary disease (COPD) is characterized by persistent respiratory symptoms and airflow limitation due to airway and/or alveolar abnormalities that develop after significant exposure to noxious particles or gases, including tobacco smoke [1]. Smoking is the most significant risk factor for COPD and pulmonary fibrosis [2]. Combined pulmonary fibrosis and emphysema (CPFE) is characterized by co-existing emphysema and pulmonary fibrosis. Cottin et al. initially described CPFE in a cohort of 61 patients with both emphysema in the upper zones and diffuse parenchymal lung disease with fibrosis in the lower zones of the lungs on chest high-resolution computed tomography (HRCT) [3]. Smoking is suggested to play a crucial role given that almost all (98%) patients with CPFE are current or former smokers [4]. It remains unclear whether co-existing emphysema and pulmonary fibrosis is a distinct clinical entity or a coincidence of two smoking-related diseases within an individual as seen in the coexistence of lung cancer and COPD. Since its proposal, CPFE has been described in the context of idiopathic pulmonary fibrosis and other chronic lung fibrotic diseases, including connective tissue-related interstitial lung diseases [5]. Interstitial lung disease with the fibrotic CPFE component is yet to be established. Tobacco smoking has been reported to possibly result in areas of increased lung density on HCRT, which are termed as interstitial lung abnormalities (ILAs). Further, HRCT scan analysis of a large cohort showed that 8% of the smokers presented ILAs [6].

We conducted a cross-sectional study to analyze prospectively data collected from a Japanese COPD registry. We evaluated the following: 1) the frequency of newly appeared ILAs during the observational period in the patients with COPD and 2) their clinical characteristics. Specifically, we compared the annualized rate of COPD exacerbations per patient during the observational period between patients with COPD with and without comorbid ILAs.

## Materials and methods

### Study design

We conducted a cross-sectional study to analyze data prospectively collected from consecutive scheduled visits or newly registered patients in the Ishinomaki COPD Network (ICON) Registry [7,8] between April 2012 and November 2018. Briefly, the ICON is a regional medical liaison system aimed at providing comprehensive care to patients with COPD and is a part of a multi-center interdisciplinary collaboration among health care providers; specifically, respiratory medicine specialists, nurse specialists, therapists, pharmacists, and general practitioners, in Ishinomaki, Japan. In accordance to the COPD statements or guidelines [9,10], patients registered to the ICON Registry receive the standard therapy and care in general practice clinics. Further, patients undergo scheduled examinations and receive education at the Japanese Red Cross Ishinomaki Hospital (a 464-bed tertiary community hospital) every 6–12 months. Patients who experience exacerbations are first treated by their general practitioners, and if necessary, are subsequently referred to the Japanese Red Cross Ishinomaki Hospital. The patient education program includes training on early exacerbation recognition and a written action plan for exacerbations using a self-management diary. Patients are prescribed short-acting bronchodilators, but not oral corticosteroids or antibiotics, for self-administration during exacerbations.

### Ethical considerations

This study is part of an ongoing COPD cohort study registered with the University Hospital Information Network Clinical Trials Registry (identifier: UMIN000017376).

All the patients provided written informed consent and the study was approved by the Ethics Committee of the Japanese Red Cross Ishinomaki Hospital (approval number: 12-14-1).

## Patients

We enrolled patients with stable COPD (age range: 40–90 years) who were former smokers with a smoking history of at least 10 pack-years. Patients suspected of COPD were referred to the Japanese Red Cross Ishinomaki Hospital. Patients who were diagnosed as COPD and confirmed as stable condition were registered to the ICON registry, if they gave their consent. All the patients were diagnosed with COPD according to Global Initiative for Chronic Obstructive Lung Disease (GOLD) criteria. Repeated spirometry was used to confirm persistent airflow limitation, which is defined as a post-bronchodilator forced expiratory volume in 1 second (FEV1)/forced vital capacity (FVC) ratio of < 0.7. The exclusion criteria were as follows: current smokers; chronic bronchitis or emphysema without airflow limitation; hematologic disease; lung resection history; use of oral corticosteroids, immunosuppressive agents, or antifibrotic agents; and COPD exacerbation occurrence within the 4 weeks preceding the data collection.

Further, in this study, we excluded patients lost to follow-up, as well as patients who did not complete the 1-year follow-up, received radiotherapy or chemotherapy within 1 year after registry, or had obvious ILAs.

## Clinical and physiologic measurements

We recorded each patient's sociodemographic characteristics, smoking status, and maintenance treatments. Also, we calculated the body mass index (BMI) in kg/m$^2$ and evaluated dyspnea using the modified Medical Research Council (mMRC) dyspnea scale [9,10].

We assessed the COPD-related health status using the COPD Assessment Test (CAT), which is an eight-item questionnaire with a possible total score of 0–40 with a higher score indicating a worse quality of life [11,12].

A well-trained technician conducted the pulmonary function tests following standard guidelines under a stable condition [13]. We classified the airflow limitation severity based on the GOLD staging as follows: GOLD 1: FEV1 ≥ 80% predicted; GOLD 2: 50% ≤ FEV1 < 80% predicted; GOLD 3: 30% ≤ FEV1 < 50% predicted; or GOLD 4: FEV1 < 30% predicted [1].

We defined exacerbations as the use of antibiotics and/or systemic corticosteroids for worsening respiratory symptoms with no evidence of an alternative diagnosis [1]. We did not consider mild exacerbations treated only with short-acting bronchodilators. We evaluated the exacerbation frequency during the observational period based on direct patient interviews, patient/caregiver-maintained diaries, referral letters from general practitioners and medical record review.

Three researchers evaluated the HRCT scans. The ILA patterns, including honeycomb-like lesions, reticular abnormalities, and ground-glass opacities (Fig 1), were evaluated according to the method by Washko Gr et al. [6]. Honeycombing refers to clustered cystic airspaces of a typically consistent diameter (3–10 mm, but occasionally larger) with thick well-defined walls. It is usually accompanied by a reticular pattern containing traction bronchiectasis and bronchiolectasis [14]. However, interstitial fibrosis adjacent to a bronchiole with minimal associated emphysema, which mimics honeycombing (Fig 1), observed on CT images of patients with COPD and comorbid ILAs is termed as honeycomb-like lesions.

## Statistical analysis

Data were shown as mean (standard deviation) with categorical data being expressed as percentages. We assessed between-group differences in the continuous variables using Student's t-test or Mann-Whitney U-test. We compared categorical variables using Fisher's exact test. We analyzed the following baseline characteristics: age, sex, BMI, FEV1, FVC, mMRC dyspnea

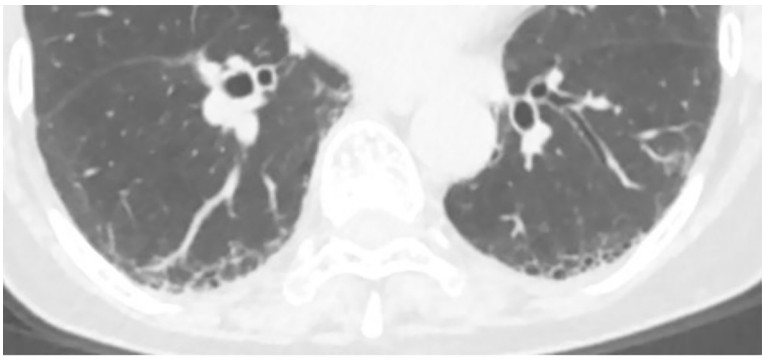

Honeycombing-like lesion

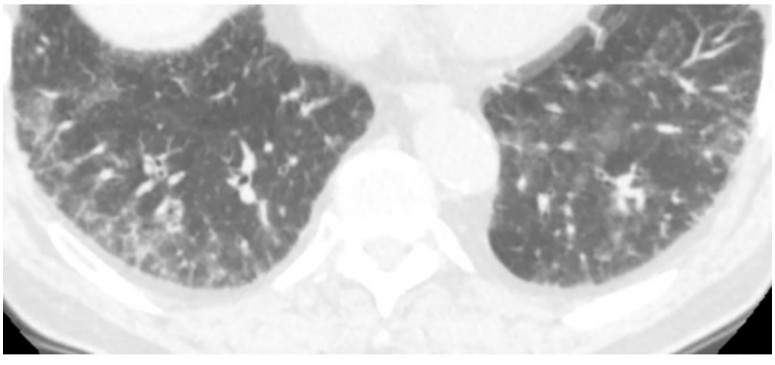

Reticular shadow, ground glass

**Fig 1. Representative pictures of the patterns of ILAs (honeycombing-like lesion, reticular abnormalities, and ground-glass opacities).**

scale score, CAT score, and the annualized rate of COPD exacerbations per patient during the observational period. To reduce selection bias, we calculated a propensity score through logistic regression with age, sex, BMI, and GOLD stage as the dependent variables [15]. We performed 1-to-3 propensity score matching between the ILA and control groups.

We performed all statistical analyses using EZR (Saitama Medical Center, Jichi Medical University, Saitama, Japan), which is a graphical user interface for R (The R Foundation for Statistical Computing, Vienna, Austria) [16]; P < 0.05 was considered as significant.

## Results

We enrolled 576 consecutive patients with COPD; among them, 463 were identified as eligible for study inclusion (Fig 2). To evaluate newly appearing ILAs and to avoid the possible effects of lung abnormal shadows, we excluded patients with obvious ILAs at the registry or patients who had undergone radiotherapy or chemotherapy within the first post-registration year (Fig 2). Table 1 presents the characteristics of the study participants. The mean age of the patients was 72.9 years (standard deviation (SD) = 7.5) with 33 of them being female. The mean

Excluding
 Chronic bronchitis or emphysema without airflow limitation
 Hematologic disease or lung resection history
 Use of oral corticosteroids, immunosuppressive agents, or antifibrotic agents
 COPD exacerbation in the 4 weeks preceding the data collection

576 patients with post-bronchodilator FEV1/FVC < 70%, aged 40 or over

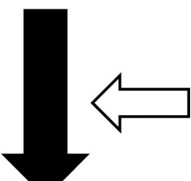

Excluding
 Loss at follow-up or did not complete 1-year-follow-up (79)
 Radiotherapy or chemotherapy within 1 year after registry (13)
 Obvious interstitial lung abnormalities (21)

463 patients with COPD included in this study

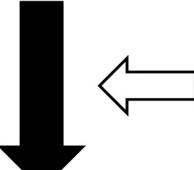

Categorized based on appearance of interstitial lung abnormalities (ILAs) during follow-up period.
The interstitial lung abnormalities on CT scan were evaluated at an ICON outpatient clinic.

30 patients with ILAs (6.5%), 433 patients without ILAs (93.5%)

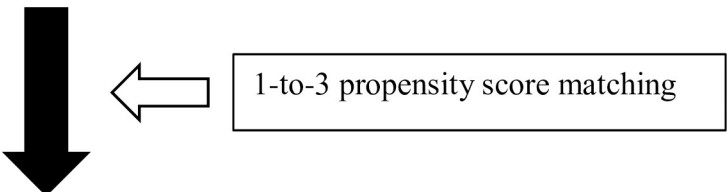

1-to-3 propensity score matching

30 patients with ILAs, 90 patients without ILAs

**Fig 2. We conducted a cross-sectional study to analyze data prospectively collected from consecutive scheduled visits or newly registered patients in the Ishinomaki COPD Network (ICON) registry between April 2012 and November 2018.**

**Table 1. Characteristics of the study patients (n = 463).**

| | | |
|---|---|---|
| Age, years | | 72.9 (7.5) |
| Female | | 33 (7) |
| BMI, kg/m2 | | 23.8 (9.2) |
| Smoking history (pack-years) | | 53.9 (29.8) |
| FEV1 (L) | | 1.62 (0.60) |
| %FEV1 (%) | | 63.1 (20.9) |
| FVC (L) | | 3.15 (0.75) |
| GOLD stage | | |
| | 1 | 106 (23) |
| | 2 | 227 (49) |
| | 3 | 102 (22) |
| | 4 | 28 (6) |
| mMRC dyspnea scale grade | | 1.0 (0.9) |
| CAT score | | 6.1 (5.4) |
| Exacerbation history¶ | | 0.17 (0.39) |
| Death from any cause§ | | 50 (10.8) |

Notes: Data are shown as mean ± SD or number (percentage).

¶ Annualized rate of COPD exacerbations during the observational period.

§Death from any cause was defined as death occurring after ICON Registry.

absolute FEV1 was 1.62 L (SD = 0.60) while the mean FEV percent predicted was 63.1% (SD = 20.9).

Among the 463 patients with COPD, newly appearing ILAs were observed in 30 (6.5%) patients (Fig 2). CT images of patients with COPD and comorbid ILAs showed honeycomb-like lesions (Fig 1).

Airflow limitation severity in patients without ILAs was distributed across GOLD 1 to 4. Contrastingly, the airflow limitation severity in patients with ILAs was distributed across GOLD 1 to 3 with a small number of patients in each class (Table 2).

To minimize between-group differences in the baseline characteristics, we matched the participants using propensity scores. The number of patients in the ILA and control groups after 1-to-3 propensity score matching was 30 and 90, respectively, which provided between-group balance in the distribution of most covariates (Table 3). After propensity score matching, the annualized rates of COPD exacerbations per patient during the observational period were 0.06 and 0.23 per year in the ILA group and control group, respectively (P = 0.043). Among the patients, two (6.7%) and 48 (11.1%) patients with and without ILAs, respectively, died from any causes; there was no significant between-group difference. No hospitalization or deaths occurred due to the rapid progression of ILAs among patients with ILAs.

Table 3 shows the mean absolute FEV1, mean FEV percent predicted, mean mMRC dyspnea scale score, and CAT scores for patients in both groups. Compared with patients without ILAs, those with ILAs showed numerically, but not significantly, greater FEV1 and FEV1 percent predicted values, as well as lower mMRC dyspnea scale and CAT scores.

## Discussion

In this study, we demonstrated the clinical phenotypes of COPD with and without comorbid ILAs in a cohort of Japanese patients with COPD. A previous retrospective study reported that ILAs were present in 8% of HRCT scans obtained from a smoker cohort [6]. Further, another

**Table 2. Characteristics of the study patients with and without ILAs before 1-to-3 propensity score matching.**

| | | ILAs (+) | ILAs (-) | P-Value |
|---|---|---|---|---|
| | | (n = 30) | (n = 433) | |
| Age, years | | 74.6 (6.4) | 72.9 (7.6) | 0.179 |
| Female | | 2 (7) | 33 (7) | 1 |
| BMI, kg/m2 | | 24.4 (3.2) | 23.8 (9.5) | 0.722 |
| Smoking history (pack-years) | | 61.2 (33.6) | 53.4 (29.5) | 0.165 |
| FEV1 (L) | | 1.74 (0.50) | 1.62 (0.61) | 0.298 |
| %FEV1 (%) | | 68.1 (19.5) | 62.8 (21.0) | 0.177 |
| FVC (L) | | 3.13 (0.63) | 3.15 (0.76) | 0.897 |
| GOLD stage | | | | |
| | 1 | 8 (27) | 98 (23) | |
| | 2 | 16 (53) | 211 (49) | |
| | 3 | 6 (20) | 96 (22) | |
| | 4 | 0 (0) | 28 (6) | |
| mMRC dyspnea scale grade | | 1.0 (1.0) | 1.0 (0.9) | 0.167 |
| CAT score | | 4.9 (4.2) | 6.2 (5.4) | 0.229 |
| Exacerbation history[¶] | | 0.06 (0.10) | 0.17 (0.39) | 0.112 |
| Death from any cause[§] | | 2 (6.7) | 48 (11.1) | 0.759 |

Notes: Data are shown as mean ± SD or number (percentage).

¶ Annualized rate of COPD exacerbations during the observational period.

§Death from any cause was defined as death occurring after ICON Registry.

study reported that 13.5% of patients with COPD present ILAs [17]. In this study, we found that approximately 6.5% of the patients with COPD presented with newly appearing ILAs during the observational period.

Exacerbations of COPD are considered important events in the management of COPD because these exacerbations have negative impacts on lung function [18], health-related quality

**Table 3. Characteristics of the study patients with and without ILAs after 1-to-3 propensity score matching.**

| | ILAs (+) | ILAs (-) | P-Value |
|---|---|---|---|
| | (n = 30) | (n = 90) | |
| Age, years | 74.6 (6.4) | 75.0 (7.1) | 0.778 |
| Female | 2 (6.7) | 8 (8.9) | 1 |
| BMI, kg/m2 | 24.4 (3.24) | 25.5 (19.4) | 0.757 |
| Smoking history (pack-years) | 61.2 (33.6) | 50.3 (29.7) | 0.096 |
| FEV1 (L) | 1.74 (0.50) | 1.58 (0.59) | 0.184 |
| %FEV1 (%) | 68.1 (19.5) | 64.0 (21.4) | 0.354 |
| FVC (L) | 3.13 (0.63) | 3.02 (0.68) | 0.442 |
| mMRC dyspnea scale grade | 0.7 (0.7) | 1.0 (0.9) | 0.09 |
| CAT score | 4.9 (4.2) | 6.4 (5.4) | 0.183 |
| Exacerbation history[¶] | 0.06 (0.10) | 0.23 (0.46) | 0.043* |
| Death from any cause[§] | 2 (6.7) | 10 (11.1) | 0.728 |

Notes: Data are shown as mean ± SD or number (percentage).

¶ Annualized rate of COPD exacerbations during the observational period.

§Death from any cause was defined as death occurring after ICON Registry.

*P < 0.05 was considered statistically significant.

of life [19,20], prognosis [21], and socioeconomic costs [22]. Previous studies reported combined pulmonary fibrosis and emphysema showed acute exacerbation of these disease [23], and ILAs were considered as one of the risk factors of exacerbations. A retrospective study on ILAs in patients with COPD reported that no patients exhibited $\geq 2$ exacerbations in one year [17]. Compared with patients without ILAs, we found a lower average annualized frequency of COPD exacerbations during the observational period in those with ILAs (0.06 vs. 0.23 per year; P = 0.043). These findings indicate that ILA development had little influence on COPD exacerbations.

Previous reports have suggested that the prognosis of patients with coexisting fibrosis and emphysema is worse than that of patients only with COPD [24,25]. However, smoking-related ILAs are different from specific forms of fibrosing lung disease associated with poor prognoses, especially usual interstitial pneumonia [26]. We did not find a significant difference in the mortality rate between-COPD and COPD followed by ILAs groups. Further, the ILAs were dominated with reticular abnormalities, ground-glass opacities, and honeycomb-like lesions. We did not observe rapid ILA progression in the patients with COPD with comorbid ILAs. This suggests that the pathogenesis of COPD with comorbid ILAs is different from that of CPFE. A previous study on lobectomy specimens of neoplasm excisions obtained from smokers demonstrated smoking-related interstitial fibrosis that could not be classified as a specific form of interstitial lung disease [26]. This lesion was characterized by varying degrees of alveolar septal widening due to collagen deposition with emphysema and respiratory bronchiolitis. The fibrosis was observed both in the subpleural and deeper parenchyma. It is difficult to differentiate emphysema from interstitial fibrosis and the clinical border between the two conditions within the same patient is vague. We speculate that ILAs that occur in patients with COPD might involve smoking-related interstitial fibrosis, which is different from specific forms of fibrosing lung disease associated with poor prognoses. Between-group analysis with adjustment for prognostic factors of COPD showed that ILAs had no adverse impact on clinical outcomes.

Previous studies have reported the occurrence of smoking-related ILAs in former smokers [27]. Tobacco smoking induces alveolar epithelial cellular senescence, which is a state of replicative arrest brought on by cellular stressors, including tobacco smoke, and results in the emergence of the senescence-associated secretory phenotype [28]. The mediators secreted by these cells include various cytokines, chemokines, matrix metalloproteinases, and growth factors involved in the pathogenesis of both emphysema and pulmonary fibrosis. Although tobacco smoking has been implicated in interstitial lung diseases, including respiratory bronchiolitis-associated interstitial lung disease and desquamative interstitial pneumonia, their radiologic features are different from those of smoking-related ILAs. The degree of smoking-induced diffuse lung diseases including lung remodeling and fibrosis varies considerably between individuals. This cannot simply be explained by cumulative tobacco exposure, implying that genetic factors interplay with smoking in the determination of the eventual disease phenotype. Currently, the underlying mechanism of exacerbation alleviation in patients with COPD with comorbid ILAs remains unclear. However, the diverse lung responses to tobacco smoke imply that there could be additional endogenous or exogenous co-factors which result in the induction of specific disease phenotypes in patients with COPD [29].

The primary strengths of our study include its prospective observational design and inclusion of community-dwelling patients treated by general practitioners in Ishinomaki or surrounding cities, which reflects the real-world COPD population in Japan. However, this study has some potential limitations. First, we cannot exclude the possibility that corticosteroid treatment for COPD exacerbations might have affected the development of ILAs. Immunosuppressive therapy has been reported to affect ILAs in patients with evidence of active inflammation,

including ground-glass opacities [4] and nonspecific interstitial pneumonia [30]. Given the low exacerbation frequency during the observational period and the short corticosteroid treatment duration, we assume that they did not significantly affect our results. Second, our sample size was smaller than that of previous large-scale studies. There is a need for a longer-term follow-up study with a larger group of patients.

## Conclusions

In conclusion, we observed slow progression of ILA lesions in patients with COPD and that the development of ILAs had no significant effect on COPD exacerbations. We suspect that ILAs occurring after COPD might involve smoking-related interstitial fibrosis, which is different from specific forms of fibrosing lung disease associated with poor prognoses.

## Acknowledgments

The authors would like to thank the staff of the Outpatient Clinic of the Japanese Red Cross Ishinomaki Hospital, Ishinomaki, Japan, for their help with data management. Further, we are grateful to the health care professionals affiliated with ICON for their kind help and cooperation with this research.

## Author Contributions

**Conceptualization:** Manabu Ono, Masaru Yanai.

**Data curation:** Manabu Ono, Seiichi Kobayashi, Masakazu Hanagama, Masatsugu Ishida, Hikari Sato, Tomonori Makiguchi, Masaru Yanai.

**Formal analysis:** Manabu Ono.

**Investigation:** Manabu Ono.

**Methodology:** Manabu Ono.

**Project administration:** Masaru Yanai.

**Software:** Manabu Ono.

**Supervision:** Seiichi Kobayashi, Masaru Yanai.

**Validation:** Masaru Yanai.

**Visualization:** Manabu Ono.

**Writing – original draft:** Manabu Ono.

**Writing – review & editing:** Seiichi Kobayashi, Masaru Yanai.

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
