## [Decision Letter · Decision Letter 0]

23 Jun 2020

PONE-D-20-07036

Clinical characteristics of Japanese patients with chronic obstructive pulmonary disease (COPD) with comorbid interstitial lung abnormalities: A cross-sectional study

PLOS ONE

Dear Dr. Onu,

Thank you for submitting your manuscript to PLOS ONE. After careful consideration, we feel that it has merit but does not fully meet PLOS ONE’s publication criteria as it currently stands. Therefore, we invite you to submit a revised version of the manuscript that addresses the points raised during the review process.

 I have received the comments of the reviewers on your manuscript. The specific comments of the reviewers are included below. Please provide point by point response in your revised manuscript.

We look forward to receiving your revised manuscript.

Kind regards,

Muhammad Adrish

Academic Editor

PLOS ONE

Journal Requirements:

2.In your Methods section, please provide additional information about the participant recruitment method and the demographic details of your participants.

Please ensure you have provided sufficient details to replicate the analyses such as:

a) the recruitment date range (month and year),

b) a description of how participants were recruited, and

c) descriptions of where participants were recruited and where the research took place.

3. Please provide a sample size and power calculation in the Methods, or discuss the reasons for not performing one before study initiation.

4. Please include additional information regarding the COPD Assessment Test questionnaire used in the study and ensure that you have provided sufficient details that others could replicate the analyses.

For instance, if you developed a questionnaire as part of this study and it is not under a copyright more restrictive than CC-BY, please include a copy, in both the original language and English, as Supporting Information.

6. Please amend your list of authors on the manuscript to ensure that each author is linked to an affiliation. Authors’ affiliations should reflect the institution where the work was done (if authors moved subsequently, you can also list the new affiliation stating “current affiliation:….” as necessary).

Reviewers' comments:

Reviewer's Responses to Questions

**Comments to the Author**

1. Is the manuscript technically sound, and do the data support the conclusions?

Reviewer #1: Yes

Reviewer #2: Partly

2. Has the statistical analysis been performed appropriately and rigorously? 

Reviewer #1: Yes

Reviewer #2: Yes

3. Have the authors made all data underlying the findings in their manuscript fully available?

Reviewer #1: Yes

Reviewer #2: No

4. Is the manuscript presented in an intelligible fashion and written in standard English?

Reviewer #1: Yes

Reviewer #2: Yes

5. Review Comments to the Author

Reviewer #1: An interesting manuscript in its field, indeed I agree with the authors, however; do they suggest a method that these lesions can be differentiated. Can we use some kind of vehicle that binds differently with each interstitial lesion?

Reviewer #2: Major points:

1. There is a need for more clarity in the paper. Are the authors trying to define the incidence of new ILA in COPD, or study its progression? They mention ‘the frequency of newly appeared ILAs during the observational period in the patients with COPD,’ but at the same time also use progression as a descriptive term. These are 2 different aspects, and it is not clear of serial CT’s/PFT’s were done to define progression. This is major aspect which needs clarification

2. They mention clinical characteristics, which is exacerbation frequency in COPD/ILA vs COPD only. What is the clinical significance of this exacerbation frequency in this context - with the low number with ILA, it is difficult to comment on this, and has limited clinical significance. In addition, the word clinical characteristics implies more clinical features, and may be too broad a term for what here is a specific analysis of exacerbations.

3. The mean age of these patients is 72.9 years, which is also the age for rising ILD prevalence. The fundamentally question is that is this an incidental abnormality arising in this elderly cohort, and is there any link at all with COPD – an epiphenomenon at best? What are the pathophysiological correlates which make such a relationship tenable, above and beyond the link with smoking irritants? Is it just part of a continuum, or is post-infectious scarring? Please comment on this.

4. The significance of these CT ILA types are not clear, is GGO or reticulation different from honeycombing? The true impact of these findings is not clear.

Minor points

1. The exacerbations were based on patient interviews, diaries, and record review, and need for drugs, and may not reflect a true exacerbation.

6. PLOS authors have the option to publish the peer review history of their article (what does this mean?). If published, this will include your full peer review and any attached files.

Reviewer #1: No

Reviewer #2: No

---

## [Author Response · Author response to Decision Letter 0]

10 Aug 2020

We would like to thank you and the reviewers for the helpful comments on the original version of our manuscript. 

We have taken all these comments into account and resubmit a revised version of our paper. 

In the responses to the comments of reviewers, we have included a point-by-point list of responses on separate pages.

---

## [Decision Letter · Decision Letter 1]

14 Sep 2020

Clinical characteristics of Japanese patients with chronic obstructive pulmonary disease (COPD) with comorbid interstitial lung abnormalities: A cross-sectional study

PONE-D-20-07036R1

Dear Dr. Onu,

We’re pleased to inform you that your manuscript has been judged scientifically suitable for publication and will be formally accepted for publication once it meets all outstanding technical requirements.

Kind regards,

Muhammad Adrish

Academic Editor

PLOS ONE

Additional Editor Comments (optional):

Reviewers' comments:

Reviewer's Responses to Questions

**Comments to the Author**

1. If the authors have adequately addressed your comments raised in a previous round of review and you feel that this manuscript is now acceptable for publication, you may indicate that here to bypass the “Comments to the Author” section, enter your conflict of interest statement in the “Confidential to Editor” section, and submit your "Accept" recommendation.

Reviewer #1: All comments have been addressed

Reviewer #2: All comments have been addressed

2. Is the manuscript technically sound, and do the data support the conclusions?

Reviewer #1: Yes

Reviewer #2: Yes

3. Has the statistical analysis been performed appropriately and rigorously? 

Reviewer #1: Yes

Reviewer #2: Yes

4. Have the authors made all data underlying the findings in their manuscript fully available?

Reviewer #1: Yes

Reviewer #2: Yes

5. Is the manuscript presented in an intelligible fashion and written in standard English?

Reviewer #1: Yes

Reviewer #2: Yes

6. Review Comments to the Author

Reviewer #1: I have no further comments, all comments were addressed. Again all comments were addressed I have no further comments

Reviewer #2: No further comments, the pertinent review questions have been addressed and corrections made in the manuscript

7. PLOS authors have the option to publish the peer review history of their article (what does this mean?). If published, this will include your full peer review and any attached files.

Reviewer #1: No

Reviewer #2: No

---

## [Editor Report · Acceptance letter]

29 Oct 2020

PONE-D-20-07036R1 

Clinical characteristics of Japanese patients with chronic obstructive pulmonary disease (COPD) with comorbid interstitial lung abnormalities: A cross-sectional study 

Dear Dr. Ono:

I'm pleased to inform you that your manuscript has been deemed suitable for publication in PLOS ONE. Congratulations! Your manuscript is now with our production department. 

Kind regards, 

on behalf of

Dr. Muhammad Adrish 

Academic Editor

PLOS ONE